# The Impact of Digital Technology Use on Passive Entrepreneurial Exit in Rural Households: Empirical Evidence from China

**Yiran Wang, Zhijian Cai * and Jie Wang**

College of Economics and Management, Nanjing Forestry University, Nanjing 210037, China
* Correspondence: janecai69@njfu.edu.cn

**Abstract:** To reduce the passive entrepreneurial exit of rural families and improve the quality of rural entrepreneurship, we theoretically discuss how digital technology can help rural families to obtain resources in the process of entrepreneurship from a micro perspective to inhibit the passive entrepreneurial exit, and perform an empirical test by using the data of three consecutive periods of China's household finance survey in 2015, 2017, and 2019. The results show that: First, digital technology use has a significant inhibitory effect on the passive entrepreneurial exit of rural families. Second, the role of digital technology lies in effectively improving the social capital, human capital, and financial capital of entrepreneurial families, thus inhibiting the passive entrepreneurial exit of rural families. Third, the role of digital technology is related to the development level of village social networks and county economic development levels. The average marginal effect of data technology in villages or counties with high development levels is higher. The policy implications are as follows: We should accelerate the improvement of rural digital infrastructure construction, accelerate the expansion of digital services for the people, and enhance the ability of farmers to use digital technology while increasing policy support for farmers' entrepreneurship.

**Keywords:** passive entrepreneurial exit; digital technology use; social capital; human capital; financial capital

## 1. Introduction

Encouraging and supporting rural innovation and entrepreneurship is one of the most important mechanisms to stabilize farmers' income [1]. However, the shortage of entrepreneurial resources is one of the critical factors limiting the development of innovative entrepreneurship among rural farmers [2]. It leads to a higher exit phenomenon in farmer entrepreneurship (Data from the China Household Finance Survey (CHFS) show that nearly 40% of the sample of rural households that operated a business or industry in 2017 will no longer operate a business or industry in 2019 due to lack of capital, unprofitability, and industry competition). The 2019 Global Entrepreneurship Monitor report also shows that the exit rate of Chinese entrepreneurs is close to 1/4 of new startups. According to Justo [3], entrepreneurial exit is either an active type of exit due to harvesting investment, and more employment opportunities, or a passive type of exit due to poor entrepreneurial performance, poor operation, and industry competition. Tian F [4], based on the analysis of the third national economic census data, found that small and micro enterprises in China show a high birth rate, high mortality rate, a shorter average life expectancy, and lower economic efficiency. The exit of farmers' entrepreneurship in China belongs to passive exit, which is essentially due to the shortage of resources, either lack of capital, lack of market resources, and technical resources, or inferior management ability in the competition. Moreover, the 2019 Global Entrepreneurship Monitor report also shows that, in general, China has a low percentage of technology-based entrepreneurship (2.66%) and the main reason for the termination of entrepreneurial activity is that the business was not profitable.

In contrast, countries such as Australia (13.1%), the UK (11.27%), and Japan (10.58%) have a high percentage of technology-based entrepreneurship, and the main reason for their entrepreneurial exit is the opportunity to sell or discover. They exit because they have the opportunity to sell or find other business opportunities. This shows that China has a higher proportion of passive exits than other countries. Due to the high percentage of China's rural population, Chinese households play a critical role in the sustainable development of the Chinese economy. The cost of too frequent business exits can be high and disruptive to the market, which is detrimental to the sustainable development of the Chinese economy. Therefore, it is a topic of interest how farmer entrepreneurs can obtain resources to match the survival of their entrepreneurship and then reduce passive exit.

Entrepreneurial exits are a shared global phenomenon. By searching the topics of "Entrepreneur*", "Founder", or "entrepreneurial exit", we found that previous literature focused on the start-up and growth phases of entrepreneurship, focusing on the identification and utilization of entrepreneurial opportunities and the acquisition and integration of resources, but less on the exit of entrepreneurship. However, the entrepreneurial process is not only limited to the start-up and growth phases but also includes the exit phase. Usually, researchers summarize the research on factors related to entrepreneurial start-up and growth of farmers from three characteristics. In terms of individual characteristics, they mostly focus on age, gender, marriage, and whether they are party members, etc. [5,6]. In terms of household characteristics, they mainly focus on household assets, the proportion of older people, etc. [7,8]. From the external characteristics mainly focused on the policy system, entrepreneurial environment, etc. [9,10], De Tienne [11] defines entrepreneurial exit as the process by which the founders of an entrepreneurial team or private firm leave the company they created and exit, varying degrees, from the decision-making structure and ownership of the firm. As with entrepreneurial entry, scholars have primarily focused on exploring the factors that influence it. They argue that the factors influencing entrepreneurial exit behavior include macro, organizational, and individual characteristics. The macro level includes economic development and institutional environment; and organizational factors include corporate resources and financial performance; individual factors include unique personality traits and human capital. He [12] explored why entrepreneurs exit from entrepreneurship from three perspectives of the institutional environment. De Tienne [13] examined the impact of strategic resources and social wealth of family businesses on entrepreneurial exit from the aspect of corporate entrepreneurship. Kato [14] and Rocha [15] studied the impact of the human capital of entrepreneurs on entrepreneurial exit.

Overall, the process of entrepreneurial exit is very complex, and scholars' explanations of the entrepreneurial exit phenomenon are still limited. Scholars have mainly studied entrepreneurial exit at the macro and organizational levels. At the micro level, they have mostly studied the individual characteristics of entrepreneurs in general, and there are fewer studies related to the entrepreneurial exit of marginalized groups-farmers. Furthermore, entrepreneurial exit is divided into the active exit and passive exit, and in the micro-level analysis, active exit and passive exit should be distinguished according to the reasons for the exit, which has been neglected in previous studies. Moreover, since entrepreneurship is a dynamic process including entrepreneurial entry and exit [16], it is necessary to explore the causes of passive exit of rural households from the perspective of the state change of "entrepreneurial-non-entrepreneurial".

According to the «Statistical Report on the Development of China's Internet», as of August 2021, the number of rural Internet users in China was 297 million, and the Internet penetration rate in rural areas was 59.2%, an increase of 3.3 percentage points from December 2020, and the proportion of administrative villages connected to fiber optics and 4G was over 99%. The use of digital technology can effectively break the space-time barrier and improve the universalization of limited resources, the essence of which is the digitization of resources, thus becoming an effective way for farmers to access resources for innovation and entrepreneurship: digital technology, represented by the Internet and smartphones, has effectively changed the form of operation and organization of traditional market participants,

creating a large number of business opportunities [17,18]. Digital technology use enhances farmers' social interactions and contributes to the acquisition of more prosperous social capital [19]. Digital technology use significantly increases household income by facilitating the development of digital finance [20]. In addition, the Internet also facilitates entrepreneurial knowledge learning as a way to improve human capital [21].

Can digital technology use also inhibit farmers' passive entrepreneurial exit by facilitating access to entrepreneurial resources? We aim to explore whether there is a link between the phenomenon of household passive entrepreneurial exit and rural households' digital technology use. In addition, we expand the capital transmission mechanism through which digital technology use affects rural households' passive entrepreneurial exit behavior. We want to discover more beneficial effects of digital technology use on economic and social development, and thus provide some suggestions for advancing quality entrepreneurship in developing countries. Possible innovations of this paper: First, we focus on analyzing the impact of digital technology use on the exit behavior of residential households hidden under the phenomenon of the entrepreneurial wave from a microscopic perspective. Second, it distinguishes that rural household entrepreneurial exits are mainly passive ones. Third, the mechanism is explored from three perspectives: social capital, human capital, and financial capital. Given that the research question is the impact of digital technology use on rural households' entrepreneurial exit, we use a nationwide survey-based database (China household finance survey) that includes complete questionnaire information on digital technology use, entrepreneurial exit, and other related information that is a better fit with the research question. And this is secondary data. This paper intends to use tools such as Stata13 and Spss to explore empirically using relevant data from the China Household Finance Survey (CHFS) for 2015, 2017, and 2019.

## 2. Theoretical Analysis

### 2.1. Why People Quit Entrepreneurship

Entrepreneurial exit is explored chiefly from a profit-maximization perspective, where poor business performance is the main reason for entrepreneurial exit [22]. For farmers who start their businesses, as mentioned earlier, farmers passively exit their businesses due to lack of capital, unprofitability, industry competition, and other reasons. Whether it is the entrepreneurial exit of general groups or teams or the entrepreneurial exit of marginal groups such as farmers, the root causes are mostly passive entrepreneurial exits due to the shortage of entrepreneurial resources to sustain the enterprise's survival. According to resource opportunity theory, the survival and development of an enterprise is a process of acquiring various resources [23]. Porter [24] pointed out in his work that entrepreneurial resources are the sum of different types of factors that are successively invested and utilized in the entrepreneurial process and are the basis for the establishment and growth of the enterprise. Porter classifies entrepreneurial resources into two categories: primary and advanced, in which asset-based resources such as capital belong to primary resources, and knowledge-based resources such as information, technology, and management personnel belong to advanced resources. With the establishment and development of enterprises, not only do they need sufficient primary resources, but the marginal contribution of advanced resources to the growth of enterprises is also gradually increasing, and the more advanced resources an enterprise has, the less likely it is to exit the market passively.

### 2.2. Digital Technology Use and Passive Entrepreneurial Exits

With the rapid development of digital technologies such as computers, smartphones, and the Internet, the penetration rate of digital technologies is high not only in urban areas but also vastly increased even in rural areas. Digital information technology access represented by the Internet has broken the boundaries of time and space, and more and more entrepreneurs are carrying out entrepreneurial activities through digital technology. During the period when the Internet was not popular, the problem of lack of resources for entrepreneurship in rural households in China was more serious compared to urban

household entrepreneurship. Entrepreneurs faced not only the constraints of primary resources such as capital, but also the inaccessibility of advanced resources such as technology, market, and business management, which led to the inability to continue entrepreneurial activities and only passive withdrawal from the business [25]. The use of digital technology has a positive impact on the acquisition of entrepreneurial resources [26]. In addition, the integration of mobile Internet with new generation digital technologies such as big data and artificial intelligence has changed the business and organizational forms of traditional market participants, such as breaking the physical barriers between urban and rural areas and extending the agricultural and rural tourism industry chain, creating a large number of business opportunities [17]. The acquisition of digital skills can help farmers gain access to a broader range of economic opportunities [18], and entrepreneurs can accumulate their own human and social capital to improve their business management capabilities with digital skills [2]. The business and professional information available through the Internet helps farmers better grasp market dynamics and improve their entrepreneurial skills [27]. The dual advantages of the market environment and technological environment that e-commerce has in the digital economy optimizes the entrepreneurial environment in rural areas [28]. Digital finance built based on information technology such as the Internet, big data, and cloud computing can significantly reduce the cost of services and increase the penetration rate of financial services by financial institutions, thus significantly enhancing the availability of financial resources, especially improving the availability of financial services and meeting the financial needs of those disadvantaged groups excluded by traditional financial institutions [29]. Therefore, this paper proposes the first hypothesis:

**Hypothesis 1.** *Digital technology use inhibits the passive entrepreneurial exit of rural households.*

*2.3. Mechanisms of the Role of Digital Technology Use on Passive Entrepreneurial Exit*

Resources are the key to entrepreneurship, and entrepreneurs' access to such social capital, human capital, and financial capital can provide the resources and values needed for business survival and growth, thereby inhibiting business exit [30]. Guo X B [31] also pointed out that the entrepreneurial resource endowment of entrepreneurs includes social capital, human capital, and financial capital, which is a combination of entrepreneurial capital. Based on this, exploring the impact of digital technology use on entrepreneurial exit from the perspective of resources should be done from three perspectives: social capital, human capital, and financial capital. Social capital refers to the resources embedded in social networks that actors acquire and use in their actions" [32]. Granovetter [33] divides social networks into relational strong social networks based on solid emotions and weak relational social networks based on "generalized friendships." The use of digital technology can increase the efficiency of communication within the strong relational social networks of rural households and thus increase the acquisition of tacit knowledge resources such as markets, industries, and environments [34]. Digital technology use can also be more advantageous for acquiring asset resources by facilitating solid relationships. Strong relationships in rural households will take higher risks and provide firms with the required asset-based resources such as financial, material, and human resources based more on an emotional perspective, thus becoming the main channel for firms to obtain resources. Weak relationship networks have apparent advantages in information and knowledge provision, providing heterogeneous and non-redundant information, a large amount of information, and a wide range of information sources. Rural households can obtain more information and thus more entrepreneurial resources through their weak ties outside their small circles. In addition, sparse networks facilitate firms to find more entrepreneurial resource providers who can provide them with more knowledge-based and asset-based resources such as financial resources production, and complementary technology distribution channels [35]. Weakly relational social networks expand by "building relational bridges", but spatial and temporal constraints limit their scope. In contrast, the use of digital technology can greatly expand the coverage of weakly connected social networks [34]; individuals

and organizations are increasingly using digital technology to build and strengthen ties with the outside world, increase the closeness of network members through formal and informal communication, and improve individuals' integration into social networks [36], thus helping rural households to establish a wider range of weakly connected social networks. Accordingly, this paper proposes a second hypothesis.

**Hypothesis 2.** *The use of digital technology increases the social capital of entrepreneurs, thereby inhibiting the passive entrepreneurial exit of rural households.*

Human capital. Entrepreneurial human capital characteristics such as education, age, work experience, and psychological factors have been considered to play a critical role in the survival and growth of a firm [37]. However, human capital is multidimensional, and the entrepreneurial knowledge and entrepreneurial skills acquired by the entrepreneur are critical for the survival and growth of the firm as opposed to the prior experience or level of education acquired by the entrepreneur [38]. Entrepreneurs should know a wider range of fields and be able to integrate this knowledge instead of a superficial level of education. Entrepreneurs should be able to explore market-valuable opportunities, access relevant information and resources through keen observation of the uncertain environment, and analyze information and integrate resources to make good use of market opportunities [39]. Therefore, the analysis of the impact of human capital on entrepreneurial survival and business development only in terms of educational background or prior work experience is incomplete. The use of digital technology enhances human capital by changing learning styles, widening access to knowledge, speeding up knowledge acquisition, and providing more affluent and more practical learning materials and information, which in turn promotes productivity growth and achieves higher wages. This "wage premium" is significantly present in disadvantaged groups where digital technology talent is scarce [40]. In reality, the more knowledge and skills rural residents learn from digital network platforms, the higher the economic returns, which is more significant in the relatively disadvantaged group of rural residents [41]. From the perspective of rural household entrepreneurship, entrepreneurs can accumulate their own human capital with digital skills and improve their business management and social activity capabilities while acquiring knowledge resources that are closely related to entrepreneurship and thus improve entrepreneurial performance [42]. For example, with the development of inclusive digital finance in recent years, the financial knowledge of rural household entrepreneurs has generally improved. Not only in terms of interest rate calculation, inflation understanding, and investment risk identification, but also rural households are more capable of interpreting relevant market development opportunities, resources, products, etc., the better they understand the business and market environment, the better they can identify, access, and utilize more and better resource opportunities [43]. On the other hand, the higher the financial literacy, the better the ability to allocate resources and opportunities, and the more likely it is that the firm's resources and opportunities will be combined in an optimal situation, and thus the better the firm operates and the lower the likelihood of a passive exit from entrepreneurship. Accordingly, this paper proposes a third hypothesis.

**Hypothesis 3.** *The use of digital technology enhances the level of the human capital of entrepreneurs, thereby inhibiting passive entrepreneurial exit.*

Financial capital is an essential prerequisite for guaranteeing the fundamental rights and interests of financial accessibility for disadvantaged groups and achieving income growth for low-income people through financial scale expansion [44]. However, it is very difficult to obtain sufficient start-up capital from the traditional financial market in rural household entrepreneurship in China due to the lack of qualified collateral. Most entrepreneurs need to rely on their preliminary capital accumulation or loans from friends and relatives [2]. Therefore, in the development process of their commercial enterprises, the primary capital factor that limits their business activities is often digital finance. Digital

finance based on digital technology effectively shortens household investment needs and the distance of financing demand and financial service supply, establishes diversified investment channels, reduces the distance of financial service supply, and provides financial information. It not only saves transaction time and costs but also alleviates the problem of information asymmetry between borrowers and lenders, lowering the threshold for individuals to enter the financial market for investment and financing [29]. Individuals with digital skills can borrow the advantages of online platform information transfer to break the spatial and information barriers and effectively alleviate financing constraints to increase entrepreneurship [45]. The development of digital finance significantly enhances the probability of entrepreneurship for the low social capital group but not for the high social capital group, i.e., the development of digital finance can reduce the dependence of rural residents on social networks [46]. Accordingly, this paper proposes the fourth hypothesis.

**Hypothesis 4.** *The use of digital technology increases the level of the financial capital of entrepreneurs, thereby inhibiting passive entrepreneurial exits.*

The model structure of the theoretical hypothesis is shown in Figure 1. We can see that digital technology use impacts the passive entrepreneurial exit of rural households and affects them through social capital, human capital, and financial capital. H1, H2, H3, and H4 labeled in the figure correspond to Hypothesis 1, Hypothesis 2, Hypothesis 3, and Hypothesis 4, respectively.

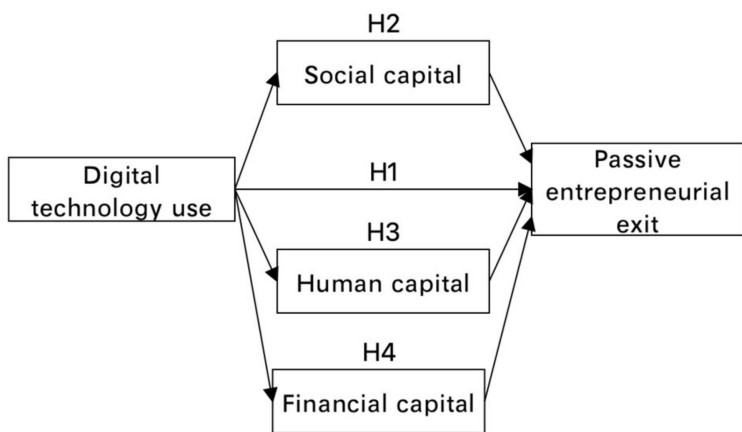

**Figure 1.** Structural model.

### 3. Data, Variables and Methods

*3.1. Data Source*

The data used in this paper come from the China Household Finance Survey (CHFS) conducted nationwide in 2015, 2017, and 2019 by the China Household Finance Survey and Research Center of Southwest University of Finance and Economics. China Household Finance Survey (CHFS) is a nationwide survey project conducted by China Household Finance Survey and Research Center, aiming to collect relevant information about Household Finance at the micro level. The main contents include housing assets and financial wealth, liabilities and credit constraints, income and consumption, social security and insurance, intergenerational transfer payments, demographic characteristics and employment, payment habits, and other related information. In order to provide high-quality micro-family financial data for academic research and government decision-making, the family economy and financial behavior are comprehensively and carefully depicted. Five surveys were successfully conducted in 2011, 2013, 2015, 2017, and 2019. The survey subjects were distributed in 29 provinces, 367 counties (districts, county-level cities), and 1481 communities. Covering 40,011 households and 127,012 individuals, it is representative of cities at the national, provincial, and sub-provincial levels.

The survey subjects were selected based on the principle of those who were engaged in business in the previous period and still participated in the survey in the latter period. Those who were engaged in business in 2015 and still participated in the survey in 2017 and those who were engaged in business in 2017 and still participated in the survey in 2019. The screening steps for the passive entrepreneurial exit subjects are: First, for entrepreneurial exits, those who were engaged in business in the previous period and still participated in the survey in the current period, that is, those who were engaged in business in 2015 and still participated in the survey in 2017 and those who were engaged in business in 2017 and still participated in the survey in 2019; then, screening passive entrepreneurial exit households from entrepreneurial exit households and according to the definition of passive entrepreneurial exits in this paper (see the definition in "Explanatory variables" below), the passive entrepreneurial subjects were selected from the entrepreneurial exit. In this way, the data from three consecutive periods form two panels combined into mixed cross-sectional data. The rural subjects were also screened to remove missing values of critical variables and invalid questionnaires, resulting in 1767 valid cross-section samples, including 739 passive entrepreneurial exit samples, 153 active entrepreneurial exit samples, and 875 entrepreneurial continuation samples. Such a screening method is reasonable.

### 3.2. Variable Selection

Explained variables: passive entrepreneurial exit. Firstly, rural households engaged in self-employment or business operation were defined as entrepreneurs, and those engaged in agricultural production and operation as well as forestry, animal husbandry, attachment, and fishing were excluded from the discussion. Then, the entrepreneurial exit was defined as the sample that was engaged in operation in the previous period but had withdrawn from operation in the latter period in consecutive surveys [38]. Since entrepreneurial exit is divided into two categories: active and passive [22], the distinction between the two is made by tracking survey data and observing the employment status of the labor force after exit to determine whether it is active or passive by using the opportunity cost comparison method to measure and differentiate according to the economic profit idea [25], which operates as follows: based on resident households' per-capita net income in the current period, if the per-capita net income of the passive entrepreneurial exit family is lower than the baseline, it is regarded as the passive entrepreneurial exit. Otherwise, it is the active exit. The value of "1" is assigned to the passive entrepreneurial exit, and "0" otherwise (including entrepreneurial continuation or active exit).

Core explanatory variables: digital technology use. According to the Statistical Report on the Development Status of the Internet in China, computers and cell phones are the main ways rural households can access the Internet in rural areas. Therefore, the use of digital technology is defined by whether the household owns a computer or a smartphone. The use of digital technology is defined when at least one of them is used. That is, when the household owns a computer or uses a cell phone that is a smartphone, the value of "1" is assigned to one of the two uses. It is "0" otherwise. The question in the questionnaire was selected: "What types of durable goods do you currently own in your household?" and "What kind of cell phone do you use" to get the indicator.

Control variables. The control variables were selected concerning existing studies on entrepreneurship to control for estimation bias at two levels: individual characteristics variables and household characteristics variables. The individual characteristics variables include age, health, gender, education level, party membership, marital status, and commercial insurance. Household characteristics include four characteristics variables: household size, whether the parents are in charge of the unit or not, fixed deposits, and household income other than business and industry. Among them, literacy and health status determine farmers' mental and physical strength, and the richer the human capital, the lower the probability that the household will exit from passive entrepreneurship. Household heads with commercial insurance indicate a greater awareness of risk and risk control, which helps to inhibit passive entrepreneurial exit. Households with parents who are unit

heads and above have a broader weakly related social network, which helps households to obtain non-redundant and heterogeneous entrepreneurial information and entrepreneurial resources. Fixed deposits and household income other than business and industry imply more prosperous household financial capital, and the lower financial constraints, the lower the likelihood of household passive entrepreneurial exit.

Descriptive statistics of the variables are shown in Table 1: 72% of rural households use digital technologies, and up to 46% of rural households passively withdraw from entrepreneurial activities.

**Table 1.** Descriptive statistics of the variables.

| Variable Name (Abbreviation) | Dummy Variable Definition | Average Value | Standard Deviation | Sample Size |
|---|---|---|---|---|
| Passive entrepreneurial exit | Passive Entrepreneurial Exit = 0, Entrepreneurial Continuity and Active Entrepreneurial Exit = 1 | 0.418 | 0.493 | 1767 |
| Digital technology use | Using a computer or smartphone = 1, otherwise = 0 | 0.720 | 0.449 | 1767 |
| Age | Unit: years | 51.339 | 11.141 | 1767 |
| Age squared | Age × age/100 | 27.597 | 11.947 | 1767 |
| Health | Normal and above = 1; otherwise 0 | 0.853 | 0.354 | 1767 |
| Gender | Male = 1, Female = 0 | 0.918 | 0.274 | 1767 |
| Education level | Unschooled = 1, PhD = 9 | 2.800 | 0.972 | 1767 |
| Party members | Yes = 1, No = 0 | 0.123 | 0.329 | 1767 |
| Marriage | Married, remarried = 1, otherwise = 0 | 0.957 | 0.202 | 1767 |
| Business insurance | With commercial insurance = 1, otherwise = 0 | 0.585 | 0.493 | 1767 |
| Family size | Number of family members | 3.602 | 2.276 | 1767 |
| Parent is the head of the unit or more | Yes = 1, No = 0 | 0.013 | 0.116 | 1767 |
| Fixed deposit | Take logarithm | 1.265 | 3.470 | 1767 |
| Income other than business and industry | Total revenue-business and industry revenue | 8.387 | 4.230 | 1767 |

### 3.3. Model Construction

Since this paper investigates the effect of digital technology use on the passive entrepreneurial exit of rural households, the explanatory variable entrepreneurial decision is a binary variable, so the Probit model is used to examine the effect and the model is as follows:

$$Pr\ (Entreprexit_i = 1) = \Phi(\alpha_0 + \alpha_1 Digital\ t_i + \alpha_2 X_i + \varepsilon_1) \tag{1}$$

In Equation (1), the subscript $i$ is the rural household sample ordinal number, $Entreprexit_i = 1$ indicates rural households with passive entrepreneurial exit, $Entreprexit_i = 0$ indicates rural households with entrepreneurial continuation or active exit, $Digital_i = 1$ indicates rural households using digital technology, and $Digital_i = 0$ indicates rural households not using digital technology. $X_i$ are the control variables, $\alpha_0, \alpha_1, \alpha_2$ are the coefficients to be estimated, and $\varepsilon_i$ is the random disturbance term. The core explanatory variable at this point is digital technology use, and the article focuses on the coefficient $\alpha_1$. If the coefficient $\alpha_1$ of the digital technology use variable is significantly negative, it indicates that digital technology use helps to inhibit passive entrepreneurial exit of rural households, validating the hypothesis proposed in the previous section.

## 4. Empirical Results and Analysis

### 4.1. Baseline Return

Baseline regression results and analysis. Estimates were performed using the Probit linear regression model in stata 13 (Stata was developed by statacorp statistical software development company in Texas, USA in 1985. It is widely used in enterprises and academic institutions worldwide. Many users work in research fields, especially in economics, sociology, politics and epidemiology), and the results are shown in Table 2. Due to the nonlinear nature of the Probit model, the results presented in Table 2 represent the average marginal effects rather than the regression coefficients. Since the model is a mixed cross-sectional model, the time effects are controlled. The results of model (1) in Table 2 show that the negative impact of digital technology on rural households' passive entrepreneurship exit is significant at the statistical level of 1%. Individual and household characteristics are gradually added to models (2) and (3), and the coefficients and significance remain robust. It is shown that digital technology use dampens the effect on passive entrepreneurial exit of rural households. Overall, digital technology has changed the business and organizational forms of traditional market participants through digital technology, creating a large number of business opportunities. Moreover, digital technology use helps rural households improve their business management capabilities and access to a large amount of business information, thus promoting business start-up continuity and inhibiting their passive start-up exit. It is clear that hypothesis 1 is tested.

**Table 2.** Model results of the baseline regression.

| Variables | (1) | (2) | (3) |
|---|---|---|---|
| Digital technology use | −0.449 ** (0.067) | −0.382 *** (0.072) | −0.389 *** (0.072) |
| Age | | −0.019 (0.019) | −0.022 (0.019) |
| Age squared | | 0.001 (0.001) | 0.001 (0.001) |
| Health | | −0.068 (0.088) | −0.0589 (0.088) |
| Gender | | 0.092 (0.114) | 0.083 (0.115) |
| Education level | | −0.089 ** (0.035) | −0.085 ** (0.035) |
| Party Members | | −0.018 (0.096) | −0.020 (0.097) |
| Marriage | | −0.203 (0.160) | −0.221 (0.159) |
| Business Insurance | | 0.080 (0.01) | −0.070 (0.102) |
| Family size | | | 0.49 *** (0.018) |
| Parent is the head of the unit or more | | | −0.042 (0.259) |
| Fixed Deposit | | | −0.024 *** (0.009) |
| Income other than business and industry | | | 0.011 (0.007) |
| Year Characteristics | Control | Control | Control |
| R-squared | 0.019 | 0.025 | 0.033 |
| Observations | 1767 | 1767 | 1767 |

Note: (i) Standard errors are in parentheses. (ii) *** and ** denote regression coefficients significant at the 1%, and 5% levels, respectively. (iii) Coefficients in the model are the average marginal effects of the variables. Same table below.

The model results also show that two of the control variables, education level and fixed deposit, have a significant negative effect on passive entrepreneurial exit in rural households, consistent with theoretical expectations. Family size has a significant positive effect, probably due to the higher support and more pressure to care for the elderly and

children in the household structure, thus leading to the passive entrepreneurial exit. All other variables are insignificant.

Endogeneity test. In identifying the causal relationship between digital technology and passive entrepreneurial exit of rural households, there may be endogeneity problems caused by measurement errors in dependent variables (the number of instruments used for digital technology makes it difficult to measure precisely) and omitted variables (not easily observable factors that affect passive entrepreneurial exit such as individual ability). Therefore, the instrumental variables approach is proposed to be used for the treatment.

Considering that digital technology usage behavior is a binary selection variable, and since IV-Probit is usually targeted to solve the problem of endogenous explanatory variables as continuous variables, using IV-Probit cannot validate and alleviate the endogeneity problem faced. CMP proposed by Roodman (2011) is introduced to validate and deal with the endogeneity problem. The CMP method can invoke various regression models, including IV-Probit. To ensure the consistency of the research methodology, the logarithm of the average village internet communication cost was finally selected as the instrumental variable for respondents' digital technology use based on several attempts of different instrumental variables. This is based on the following considerations: The digital technology use in rural households involves network communication costs, and the two are correlated; at the same time, the logarithm of the village's average network communication costs is not directly related to the household's choice of passive entrepreneurial exit, and thus satisfies the exogeneity condition.

The CMP estimation process was divided into two stages: The explanatory variable in the first stage was the use of digital technology use or not, and additional instrumental variables were added to the control variables. The first stage regression results show that the average network communication cost in villages is highly correlated with individual digital technology use. The LR test of the model estimation rejects the null hypothesis, ruling out the possibility of weak instrumental variables. The second-stage explanatory variable is the passive entrepreneurial exit, and according to the regression results in Table 3, it can be seen that the first- and second-stage regressions are significantly unchanged, and the endogenous test parameter atanhrho_12 is significant. The explanatory variable is significantly exogenous, indicating likely results using the CMP test. The inhibitory effect of digital technology use on passive entrepreneurial exit is significant at the 1% level, and the coefficient is significantly larger compared to the baseline regression results, indicating that the inhibitory effect of digital technology use on the passive entrepreneurial exit of rural households is underestimated if endogeneity is not considered.

**Table 3.** Model results of CMP regression.

| Variables | (1) | (2) | (3) |
|---|---|---|---|
| | **Explained Variables of the First Stage Model: Whether to Use Digital Technology** | | |
| Average network communication costs in villages | 0.177 *** (0.026) | 0.167 *** (0.025) | 0.167 *** (0.025) |
| Householder characteristics | | Control | Control |
| Family Characteristics | | | Control |
| Variables | Explained variables of the second stage model: passive entrepreneurial exit | | |
| Digital technology use | −1.163 *** (0.229) | −1.092 *** (0.261) | −1.147 *** (0.262) |
| Householder characteristics | | Control | Control |
| Family Characteristics | | | Control |
| atanhrho_12 | 0.482 ** | 0.346 *** | 0.373 *** |
| R-squared | 0.019 | 0.148 | 0.151 |
| Observations | 1767 | 1767 | 1767 |

Note: (i) Standard errors are in parentheses. (ii) *** and ** denote regression coefficients significant at the 1%, and 5% levels, respectively. (iii) Coefficients in the model are the average marginal effects of the variables. Same table below.

### 4.2. Robustness Tests

Replacement of estimation method. The effect of digital technology use on the passive entrepreneurial exit of rural households was estimated using the Probit model in the

previous paper. Now the robustness of the results is verified using the Logit model. The estimation results are shown in columns (1–3) of Table 4: Digital technology use on the passive entrepreneurial exit of farmers is significant at a 1% statistical level, which verifies the robustness of the previous results.

**Table 4.** Model results of robustness tests.

| Variables | (1) | (2) | (3) | (4) | (5) | (6) |
|---|---|---|---|---|---|---|
| Digital technology use | −0.726 *** | −0.610 *** | −0.618 *** | −0.414 *** | −0.324 *** | −0.332 *** |
| | (0.107) | (0.116) | (0.116) | (0.076) | (0.082) | (0.083) |
| Householder characteristics | | Control | Control | | Control | Control |
| Family Characteristics | | | Control | | | Control |
| R-squared | 0.019 | 0.025 | 0.032 | 0.016 | 0.025 | 0.031 |
| Observations | 1767 | 1767 | 1767 | 1328 | 1328 | 1328 |

Note: (i) Standard errors are in parentheses. (ii) *** denote regression coefficients significant at the 1% levels, respectively. (iii) Coefficients in the model are the average marginal effects of the variables. Same table below.

Excluding areas with advanced Internet. Considering the rural areas with more advanced Internet development in some regions, whether rural residents choose to use digital technology will be influenced by regional factors rather than a decision made entirely by farmers after their own choice, which may lead to bias in the estimation results. According to data from the Ministry of Industry and Information Technology's "Operation of Internet and Related Services in 2021", the top five provinces with high levels of Internet business development in China are Guangdong, Beijing, Shanghai, Zhejiang, and Fujian. Therefore, an attempt was made to exclude these regions before conducting the regression test. The regression results are shown in columns (4–6) of Table 4: The use of digital technology under the subsample has a significant effect on the entry and exit of farmers' entrepreneurship at the 1% level, and the regression coefficients decrease logically due to the exclusion of developed Internet regions, which again verifies the robustness of the benchmark regression.

When studying the impact of digital technology use on rural households' entrepreneurial exit behavior, it is observed that entrepreneurial exit behavior presupposes that households are engaged in entrepreneurial activity and that the group engaged in entrepreneurial activity may itself have specific characteristics. Excluding resident households without entrepreneurial activity from the sample would ignore the problem of sample selection bias due to many unobservable factors. This paper deals with the sample selection bias problem from a full-sample perspective by using the Heckman model, also known as the two-step approach, where the first step is the selection equation and the second step is the exit equation. Table 5 shows the results of the Heckman model treatment. The likelihood ratio test rejects the original hypothesis, and this result indicates that the sample selection model chosen in this paper is reasonable, and the sample does have some selection bias problems. The unbiased estimation results of the exit equation show that digital technology use has a negative effect on the passive entrepreneurial exit of rural households.

**Table 5.** Heckman model processing results.

| Variables | Heckman Model | |
|---|---|---|
| | Select Equation (to Enter or Not) | Exit Equation (Whether to Exit) |
| Digital technology use | - | −0.156 *** |
| | | (0.029) |
| Individual Characteristics | Control | Control |
| Family Characteristics | Control | Control |
| Observations | 17,405 | 1767 |
| Insigma | −0.701 | |
| Sigma | 0.493 | |
| Prob > chi2 | 55.35 *** | |

Note: (i) Standard errors are in parentheses. (ii) *** denote regression coefficients significant at the 1% levels, respectively. (iii) Coefficients in the model are the average marginal effects of the variables. Same table below.

### 4.3. Mechanism Testing

Social capital. The total income and expenditure of human gifts and the sum of annual network communication costs were selected as proxy variables for social capital for the following reasons: the income and expenditure of gifts for holidays, and red and white ceremonies are still the most important means of maintaining relationships and contacts with relatives, friends, colleagues, and customers in rural social networks; with the popularity of digital technologies such as cell phones and computers, communication and Internet access have become important ways of communicating with network members. The estimation results of social networks as mediators are shown in columns (1) and (2) of Table 6: Digital technology use positively affects social capital at the 1% significance level. In other words, digital technology can expand the strong relationship network based on relatives and family members, and the weak relationship network based on friends and colleagues to obtain the required asset-based and knowledge-based entrepreneurial resources. Digital technology use has improved the social capital of rural household entrepreneurs, and hypothesis 2 is tested.

**Table 6.** Model results of the mechanism test.

| Variables | (1) | (2) | (3) | (4) | (5) | (6) |
|---|---|---|---|---|---|---|
| Digital technology use | 1.033 *** (0.148) | 0.611 *** (0.148) | 0.321 *** (0.031) | 0.262 *** (0.034) | 1.341 *** (0.219) | 0.851 *** (0.238) |
| Control variables | | Control | | Control | | Control |
| R-squared | 0.033 | 0.129 | 0.051 | 0.093 | 0.021 | 0.050 |
| Observations | 1767 | 1767 | 1767 | 1767 | 1767 | 1767 |

Note: (i) Standard errors are in parentheses. (ii) *** denote regression coefficients significant at the 1% levels, respectively. (iii) Coefficients in the model are the average marginal effects of the variables. Same table below.

Human capital. Human capital is a critical factor that drives entrepreneurial activity, and the higher the human capital of the household head, the greater the likelihood of entrepreneurial continuation. Financial knowledge is one of the essential knowledge skills that rural residents learn from digital network platforms. Financial knowledge can help rural households better access financial markets, reduce the riskiness of investments, and provide multiple loan channels. Moreover, financial knowledge as essential human capital profoundly impacts entrepreneurship, so financial knowledge is selected as a proxy variable for human capital. The following three questions measured respondents' financial literacy: First, interest rate calculation ability, "Suppose you have $100 now, and the bank's annual interest rate is 4%. If you deposit this

USD 100 in a fixed term for 5 years, how much will you get in principal and interest after 5 years?" Second is the understanding of inflation, "Suppose you have 100 Yuan. The bank interest rate is now 5%, the inflation rate is 3% per year, what will you be able to buy after a year with your 100 Yuan?" The third is investment risk identification, "Do you think it is generally riskier to buy a company's stock alone than to buy a stock fund?". The measure of financial literacy consists of two levels: the ability to understand relevant financial and economic concepts and the possession of basic computational skills, so two dummy variables are constructed separately for each of the above questions. The first dummy variable is whether or not to answer correctly, which measures the respondents' calculation ability and takes the value of 1 if the calculation is correct, otherwise it takes the value of 0. The second dummy variable is whether or not to understand the question, which examines the respondents' understanding of financial and economic concepts, and takes the value of 1 if the respondents understand the question (including correct or incorrect calculation), otherwise it takes the value of 0 (the respondents answer that they do not know/cannot calculate) [35]. For these six variables, one public factor with a characteristic root greater than 1 was extracted using factor analysis and named financial knowledge, and the cumulative variance contribution of the rotated public factor reached 71.127%. The factor loadings of all six variables were high. The regression of the extracted common factor of financial knowledge was performed, and the results are shown in columns (3,4) of Table 6: Digital technology use positively affects financial knowledge at the 1% significance

level. This suggests that the use of digital technology promotes entrepreneurial survival, i.e., inhibits passive entrepreneurial exits. By improving rural households' financial literacy and thus resulting in better understanding and calculation of finance-related concepts, this helps them to better grasp market opportunities, identify and exploit more resource opportunities, and combine business resources with opportunities at the optimal time. Hypothesis 3 was tested.

Financial capital. According to theoretical analysis, the impact of digital technology use on financial capital is as follows: First, it directly increases financial capital; second, it relaxes the constraint of insufficient financial capital and increases lending. Credit alone cannot reflect the above logic and can be used for non-entrepreneurial purposes. Rural households may borrow for children's education, medical care, or marriage, so using the overall level of household credit does not directly reflect the issue of financial capital for entrepreneurship. So, it is proposed to use the level of household investment in entrepreneurship to reflect financial capital. The question in the Chinese household finance questionnaire is "What was the total investment at the time of participation in the project?" The estimated results are shown in columns (5,6) of Table 6: Digital technology use has a significant positive effect on financial capital at the 1% statistical level. Overall, the use of digital technology can finance rural family operations and promote the efficient development of rural network lending. Instead of relying on traditional financing methods, rural development continues to evolve with the times, using current science and technology and computer systems to carry out multimedia network financing, broadening rural financing channels. It helps rural families solve the problem of capital limitations in their business ventures. Hypothesis 4 was tested.

## 5. Discussion

The level of development of village social network. In addition to family social networks, village social networks are another critical medium and carrier of farmers' social capital. Village social networks are often seen as an essential guarantee for farmers to obtain resources to maintain daily production order in traditional agricultural societies. Even in the process of rural economic development, rural family entrepreneurs often obtain resources by participating in the construction of new village social networks and embedding in the market system. Trust is regarded as the core element of social capital. Compared with trust among members of solid relationships, trust among members of weak relationships can provide more heterogeneous and non-redundant information to transmit more tacit knowledge-based resources and provide financial and human resources to help business operation, development, and transformation [27]. Thus, it has a more prominent role in entrepreneurial maintenance. Therefore, the weak trust variable is used to measure the level of social network development at the village level.

Using the questionnaire "How much do you trust people you do not know?" as a measure of weak trust, the five choices from "very distrustful" to "very trustful" were assigned a value of "1" to "5". (There is no village code in the 2015 questionnaire, the matching rate is about 72% by matching the village code of 2017 to 2015. Because the users interviewed for the indicator "trust" in the 2015 questionnaire were not comprehensive, 473 samples were missing after screening and matching). The mean value of households within the same code was calculated according to the village indicator code as the village social network development level. Villages were divided into two groups of high and low levels with the average level of social network development as the benchmark. The model results are shown in columns (1,2) of Table 7: The marginal inhibitory effect of digital technology use on the passive exit of farmers' household entrepreneurship in villages with high social network development levels was 0.520, which is significantly higher than the effect on household passive entrepreneurial exit in low level villages, and the results are consistent with the theory.

**Table 7.** Results of heterogeneity analysis.

| Variables | (1) | (2) | (3) | (4) |
|---|---|---|---|---|
| | Low | High | Low | High |
| Digital technology use | −0.347 *** | −0.520 *** | −0.328 *** | −0.435 *** |
| | (0.119) | (0.129) | (0.097) | (0.112) |
| Control variables | Control | Control | Control | Control |
| R-squared | 0.033 | 0.049 | 0.0391 | 0.0355 |
| Observations | 698 | 596 | 881 | 886 |

Note: (i) Standard errors are in parentheses. (ii) *** denote regression coefficients significant at the 1% levels, respectively. (iii) Coefficients in the model are the average marginal effects of the variables. Same table below.

The level of economic development of the county. In the traditional society or the early stage of market economy development, the market scale is small, the market mechanism is imperfect, and the social network plays a dominant role allocating production factors. However, the growth of market forces based on the principle of efficiency will eventually reduce the role of social networks in resource allocation [47]. As China's rural transformation continues to deepen, the market also gradually becomes the primary way of resource allocation. Therefore, the market in the process of rural household entrepreneurship also becomes an increasingly important channel for entrepreneurial resources acquisition. Due to the different levels of economic development in various regions, the degree of marketization is also different. In combination with the current reality that the transformation of China's rural industrial structure from primary to secondary and tertiary industries leads to deepening non-farming, the household non-farming income is used here to represent the level of economic development as follows: Due to the sizeable extreme difference of farmers' household non-farming income data, it is more accurate to choose the median grouping than the mean grouping. Therefore, according to the median household, non-farm income within the same code is calculated as a measure of the county's economic development level, and counties with income in the top 50% are in the high economic development level group, and vice versa are in the low development level group. Group regression was performed for the two groups with high and low levels; the model results are shown in columns (3,4) of Table 7: The inhibitory effect of digital technology use on the passive exit of rural households in all counties is statistically significant at the 1% level, but the marginal impact coefficient is higher in the high level group than in the low level group, which may be because high development level counties have higher marketization and market accessibility, and rural households are more likely to obtain the resources needed for the continuation of entrepreneurship and thus discourage passive entrepreneurial exit. The results are consistent with the theory.

## 6. Conclusions

The above study leads to the following conclusions: First, digital technology use has a significant inhibitory effect on the passive entrepreneurial exit among rural households. In addition, the robustness of the replacement estimation method and the exclusion of the developed Internet region verifies that digital technology use has a significant negative effect. Second, the mechanism by which digital technology use inhibits passive entrepreneurial exit of rural households is that digital technology use improves the social capital, human capital, and financial capital. Thirdly, the marginal effect of digital technology use in inhibiting passive entrepreneurial exit of rural households is related to the level of development of village-level social networks and county economic development. The higher the level of development, the more significant the marginal effect. The use of digital technology has effectively changed the traditional forms of rural family operations and organization, stabilized and developed markets, helped to perpetuate the entrepreneurial activities of Chinese rural families, and ensured the sustainable development of the Chinese economy.

The findings of this paper are essential for further understanding the relationship between digital technology use and entrepreneurial exit behavior. The theoretical contributions of the study are as follows: First, unlike other scholars who take ordinary

entrepreneurs, start-ups or industries, and the macro environment as the object of analysis, this paper is the first to focus its research perspective on household entrepreneurship, a socially marginalized group. Based on the rapid development of digital technologies such as big data and cloud computing, this paper constructed a model of entrepreneurial exit mechanism from the perspective of digital technology use and not only analyzed its impact on entrepreneurial exit in the context of contemporary digital technologies but also distinguished the types of entrepreneurial exit and determined that entrepreneurial exit of Chinese rural households is mainly passive, which enriches the research results on entrepreneurial exit to a certain extent. Janson and Wrycza (1999) [48] analyzed the role of information technology on entrepreneurial activity using data from Poland and found that the development of information technology not only facilitated entrepreneurial activity but also contributed to the success of business operations. Using data from the UK, Mostafa et al. (2005) [49] found that the development of more entrepreneurially oriented firms will be more dependent on the Internet and that these firms perform better in exporting when they use the Internet as a medium for trade. Batjargal (2007) [50] investigated the effect of the interaction between entrepreneurs' social capital and human capital on entrepreneurial firm performance and found that the interaction between entrepreneurs' social capital and Western country experience played a positive role in the survival of entrepreneurial firms, while the interaction term between social capital and entrepreneurial experience played a negative role. However, previous studies have hardly explored the relationship between digital technology and entrepreneurial exit; therefore, this paper enriches this part of the study.

The research implications of this paper are described in terms of both theoretical and practical implications. In terms of theoretical significance, this paper explored the impact of digital technology on the passive entrepreneurial exit of rural households, distinguished entrepreneurial exit into the active entrepreneurial exit and passive entrepreneurial exit, and determined that rural households in China mainly belong to passive entrepreneurial exit. The impact of digital technology on passive entrepreneurial exit was explored from three mechanisms: social capital, human capital, and financial capital, which enriches the research on the entrepreneurial exit of rural households. In terms of practical implications, our findings have important implications for government work. With the rapid development of digital technologies, it is crucial to focus on the relationship between the use of digital technologies and the phenomenon of entrepreneurial exit to maintain a high quality of entrepreneurship. To promote the digital transformation and upgrading of rural infrastructure, the government needs to accelerate the improvement of rural digital infrastructure construction, vigorously develop digital village construction, achieve full coverage of fiber optics and 4G from the whole country to administrative villages, and increase the popularization of more cutting-edge digital technologies such as gigabit optical network and 5G. The government can also develop Internet + education so that more farmers can learn relevant entrepreneurial skills, build a digital investment and financing platform that meets the characteristics of rural entrepreneurship, and improve the accessibility of inclusive digital lending for entrepreneurial farmers. In addition, the government can provide targeted "Internet literacy" through smartphone and computer training courses. The government can also help rural "Internet novices" solve problems in the process of accessing the Internet by giving full play to the role of staff at e-commerce service stations, university student village officials, village workers, and volunteers. By building a platform to improve farmers' digital skills and skills training and by guiding enterprises and public welfare organizations to participate in farmers' digital skills such as cell phone applications, e-commerce sales, and new media applications such as Jitterbug live streaming, these initiatives not only enable farmers to apply one or more digital technologies, but also to apply them to the whole process of entrepreneurship. Finally, the government needs to increase policy support for farmers' entrepreneurship, especially for online entrepreneurship, such as agricultural e-commerce and live-streaming with

goods. Consequently, this marginal group of entrepreneurs can become an essential pillar of China's economic development transformation.

This paper has room for future expansion. It concludes that digital technology use can inhibit farmers' passive entrepreneurial exit by facilitating access to entrepreneurial resources. However, the process of entrepreneurial exit is very complex. We do not discuss this issue based on process theory but consider entrepreneurial exit as a decision-making behavior, which makes it challenging to discuss the process of household entrepreneurial exit. In the future, we can further combine "entrepreneurship-non-entrepreneurship" (entrepreneurial exit) or "non-entrepreneurship-entrepreneurship" (entrepreneurial entry) into a dynamic perspective. Moreover, most previous studies have focused on how digital technologies increase farmers' entrepreneurial resources. However, in the rural context, farmers' entrepreneurial resources are mainly obtained from social networks and markets. Previous studies have lacked discussion on how digital technologies change rural markets and social networks and thus affect farmers' entrepreneurial resources. In addition, studies on farmers' household entrepreneurial resource needs are mainly conducted from a static perspective, without considering the dynamic changes and growth characteristics of farmers' resource needs at different stages of entrepreneurship. Therefore, our team intends to draw on theories related to the entrepreneurial process and the dynamic change of entrepreneurial resource demand and explore how digital technology changes rural markets and social networks and triggers changes in farmers' entrepreneurial resource acquisition in the context of rural transformation and development in China, which ultimately leads to farmers' "non-entrepreneurship" (entrepreneurial entry) or "entrepreneurship-non-entrepreneurship" (entrepreneurial exit).

**Author Contributions:** Conceptualization, Y.W.; Data curation, Y.W.; Formal analysis, Y.W.; Methodology, Y.W. and J.W.; Software, Y.W.; Supervision, Z.C.; Writing–original draft, Y.W.; Writing–review & editing, Z.C. and J.W. All authors have read and agreed to the published version of the manuscript.

**Funding:** This research was funded by the Humanities and Social Sciences Research Planning Fund Project of Ministry of Education (Grant Number 17YJA790004) and the Key Project of Philosophy and Social Science Research in Jiangsu Universities (Grant Number 2018SJZDI099).

**Institutional Review Board Statement:** Not applicable.

**Informed Consent Statement:** Not applicable.

**Data Availability Statement:** The data presented in this research are available on request from the corresponding author. In addition, the data is provided by the China Household Finance Survey (CHFS) of Southwest University of Finance and Economics and more data can be downloaded by registering at the website. (Website: https://chfs.swufe.edu.cn (accessed on 8 July 2022)).

**Conflicts of Interest:** The authors declare no conflict of interest. All tables in the paper were prepared by the authors, and we confirm that we did not directly copy any sources.

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
