# Peer review of "The Impact of Digital Technology Use on Passive Entrepreneurial Exit in Rural Households: Empirical Evidence from China"

_sustainability, doi:10.3390/su141710662_

Round 1
Reviewer 1 Report
Congratulations on your work. I found it interesting, and I think it is based on an attractive topic. While I see these strengths, I also think there are some areas for improvement. Here are some suggestions if you would like to take them into consideration.
In my opinion, there are two fundamental problems with the paper: first, there is no review of the state of the art to see what previous work can be related to the current paper. Second, there is formally no discussion of the results (i.e., a comparison with other studies on this topic), beyond an assessment by the author(s) that they are "consistent with theory" (pg. 12).
Other questions that concern me are the following:
a) Please explain why you think it is of interest to carry out this study in this territorial context.
b) Likewise, please justify why you have chosen this sector of activity.
c) In general terms, I think that the hypotheses are poorly argued.
d) I also think that there is room for improvement in the explanation of the control variables and, more specifically, the assumptions you make about your choice (pg. 6).
e) I think you should review the wording of your results, and relate them to your hypotheses. Not only should they be "tested", you should explain whether they are confirmed or not.
f) It would be interesting to make a summary of the hypotheses that are accepted and rejected.
I hope these comments are helpful and I wish you good luck with the review.
Reviewer 2 Report
First of all, let me say that I appreciated to have the chance to review your paper. Congratulations to the authors for their interesting article! The article has considerable potential to contribute to the analysis of Entrepreneurial behavior, how it relates to Digital Technology.
The empirical section is really precise and very beneficial for all in the academic environment.
In order to help authors improve the level of their article, the following recommendations can be defined:
· The paper could be reorganized for a more fluidization of the content.
· I believe that the research part should be separated from the current state of research or that Theoretical Background.
· The section Discussion, Implications and Conclusion has to be divided into 2 parts: Section of Discussion; Section of Conclusion
· The part dedicated to the research to be completed with the exposition of the research objectives and the clear research methodology used and then followed by the formulation of the hypotheses.
· The conclusions can be accompanied by a series of implications of the study carried out by the team.
Reviewer 3 Report
1. Shortening the title is preferable – e.g. Impact of Digital Technology Use on Passive Entrepreneurial Exits of Rural Households. An Empirical Study Based on Mixed Cross-Sectional Data from China (or alike).
2. It is recommended to replace the footnotes – when the case (row 37) – by proper references, directly.
3. When making neat statements (e.g. rows 45-46), it is strongly recommended to include appropriate references next to general wording – as “scholars” (row 45); “Most of the literature” (row 55); etc.
4. It is highly recommended to display the structural model of the research – both as a figure detailed explanations – related to the research hypotheses.
5. A series of questions must be addressed, in a logical chain: (i) Which is the research objective; (ii) Which are the research questions? (iii) How the questionnaire used is related to the research questions?
6. As methodology: It is recommended to the author/s to distinctly present the research circumstances and research process – as far as secondary research and primary research.
7. What method was used for primary research? Was it survey-type?
8. If probably so (because the word “question/s” is used within section 4), it is strongly recommended to present explanations and details about the questionnaire used as research tool. It is suggested to exhibit the questionnaire (as an Annex is a possibility).
9. As far as sampling: it is not clear what “total sample” (row 222) means. Is it “total” population or just a “sample”? If the second is the case, then the selection criteria and process must be exposed, and the issue of sample representativeness must be addressed.
10. Relative to sub-section 3.1 (Data source): this section should be revised as far as use of the term “sample”; e.g. does it make sense to discuss “sample of samples”? (rows 227-229).
11. The significance of the letters used in equation (1) should be presented.
12. The Probit model is almost a century-old. Does it bring any novelty?
13. What software was used for data processing?
14. Ultimately, which are the original elements brought by the proposed paper – as compared to the state-of-the-art international literature?
15. As a matter of contribution, it is advisable to compare the research methodology and results with the data reported in international state-of-the-art literature.
16. It is recommended to relate the research results obtained in China (passive entrepreneurial exit rates of rural households) with analogue rates in other countries, and to discuss the issues observed.
17. It is suggested to discuss the research implications and present recommendations for the main stakeholders involved in the sector surveyed.
18. Are there any research limitations identified? Are there any further research possibilities foreseen? These issues should be addressed.
19. It is strongly recommended to split the last section (5.Conclusion and Policy implications) into two, better structured parts, in more concrete (less general) terms: Research implications; Conclusion.
20. Are there any sustainability issues addressed? They should be underlined.
21. It is recommended to avoid the use of personal style (“we should ... etc.”). Also considering that not author/s are the policy makers (per “policy implications”).
22. English language should be revised by a native English speaker in cooperation with author/s – so that author/s ideas and line of thinking to be accurately displayed in right English.
23. The use of punctuation marks should be revisited as well.
24. Repeatedly (row 93, row 96), Porter (2002) source is erroneously cited as “Poter”. This issue should be addressed.
Round 2
Reviewer 1 Report
Thank you for taking my suggestions into account. I think the work is clearly improved. I wish you good luck with the process.
Author Response
We have refined the paper again.
Reviewer 3 Report
(i) [Former #1] The title was shortened. Still CHFS is kept in the title and its explanation comes later – a footnote at p.2. As the journal targets international audience, the explanation of the CHFS acronym should be provided at first opportunity use.
(ii) [Former #4] Figure 1 was inserted as suggested. However, hypotheses are not pictured, and explanations are missing. Also “Passive entrepreneurial” formulation looks incomplete. In addition, each of hypotheses 2-to-4 contains two parts (statements). It is suggested to address this issue.
(iii) [Former #5] Issues were addressed partly; the last suggestion/issue stands still: to explicitly show how the quiz structure is related to the research questions.
(iv) [Former #6] As methodology: It is recommended to the author/s to distinctly present the research circumstances and research process – as far as secondary research and primary research. This recommendation stands still.
(v) [Former #7 & #8] The authors acknowledge – indirectly – the use of survey as method and questionnaire as its associated instrument. Apparently, there is no contribution as questionnaire design and development. However, the recommendation [Former #8] stands still. It is suggested to exhibit the questionnaire (as set of questions); an Annex is a possibility.
(vi) [Former #9] The issue of sample representativeness stands still; it should be discussed.
(vii) [Former #10] The confusion while using the word “sample” persists (e.g. number of “samples” used as number of instances/cases – rows 291-294). It is strongly recommended to revise the use of the word “sample” across the paper (in the framework of survey method, properly). Otherwise, when improperly used, the word should be replaced, case by case.
(viii) [Former #14 & #15] Not fully addressed. The original elements of the proposed paper (as model, methodology, findings, etc.) as compared to the state-of-the-art international literature should be neatly displayed.
(ix) [Former #16] The issue stands still: It is recommended to relate the research results obtained in China (passive entrepreneurial exit rates of rural households) with analogue rates in other countries, and to discuss the issues observed.
(x) [Former #19] Amid its development (a couple of paragraphs added) it is still suggested to split the last section (6.Conclusion) into two, better structured parts, in more concrete terms (linked to the research findings): Research implications; Conclusion.
Author Response
I have merged and uploaded the responses with the questionnaires

Round 3
Reviewer 3 Report
The proposed paper was fairly improved. However, the author/s have exaggerated while providing a copy of the "CHFS - CAPI Questionnaire (2017)"; regretfully, they missed the point ...
Overall, their work is to be appreciated.